

# The single-species metagenome: subtyping *Staphylococcus aureus* core genome sequences from shotgun metagenomic data

Sandeep J. Joseph[1,2], Ben Li[3], Robert A. Petit III[1], Zhaohui S. Qin[3], Lyndsey Darrow[2] and Timothy D. Read[1,4]

[1] Department of Medicine, Division of Infectious Diseases, Emory University School of Medicine, Atlanta, GA, USA
[2] Department of Epidemiology, Rollins School of Public Health, Emory University, Atlanta, GA, USA
[3] Department of Biostatistics and Bioinformatics, Rollins School of Public Health, Emory University, Atlanta, GA, USA
[4] Department of Human Genetics, Emory University School of Medicine, Atlanta, GA, USA

## ABSTRACT

In this study we developed a genome-based method for detecting *Staphylococcus aureus* subtypes from metagenome shotgun sequence data. We used a binomial mixture model and the coverage counts at >100,000 known *S. aureus* SNP (single nucleotide polymorphism) sites derived from prior comparative genomic analysis to estimate the proportion of 40 subtypes in metagenome samples. We were able to obtain >87% sensitivity and >94% specificity at 0.025X coverage for *S. aureus*. We found that 321 and 149 metagenome samples from the Human Microbiome Project and metaSUB analysis of the New York City subway, respectively, contained *S. aureus* at genome coverage >0.025. In both projects, CC8 and CC30 were the most common *S. aureus* clonal complexes encountered. We found evidence that the subtype composition at different body sites of the same individual were more similar than random sampling and more limited evidence that certain body sites were enriched for particular subtypes. One surprising finding was the apparent high frequency of CC398, a lineage often associated with livestock, in samples from the tongue dorsum. Epidemiologic analysis of the HMP subject population suggested that high BMI (body mass index) and health insurance are possibly associated with *S. aureus* carriage but there was limited power to identify factors linked to carriage of even the most common subtype. In the NYC subway data, we found a small signal of geographic distance affecting subtype clustering but other unknown factors influence taxonomic distribution of the species around the city.

## INTRODUCTION

Bacterial species are commonly comprised of multiple phylogenetic clades that have distinctive phenotypic properties. The process of identifying which clade a bacterial strain belongs in goes by several names but here we will refer to it as *subtyping*. Commonly used subtyping methods include multilocus sequence typing (MLST), pulsed-field gel electrophoresis (PFGE), oligotyping and variable-number of tandem-repeat typing (VNTR)

Corresponding authors
Sandeep J. Joseph,
sandeepjoseph@emory.edu
Timothy D. Read, tread@emory.edu

(*Joseph & Read, 2010*). Each of these methods was developed for bacteria first isolated in pure culture in the laboratory before DNA extraction. For early disease diagnosis of pathogenic bacterial species and to understand bacteria in the context of their natural community, it would be advantageous to subtype directly from clinical specimens such as blood and sputum. However, current direct identification options such as 16S rRNA gene sequence, FISH and REP-PCR, are not able to subtype bacteria below the species level taxonomic resolution, nor to deal with mixtures of subtypes of the same species being present.

Over the past few years, the expansion of 'metagenomic' culture-free shotgun sequencing of clinical and environmental DNA, exemplified by the human microbiome project (HMP), has generated a plethora of revolutionary new analysis methods. This includes a set of tools that has been developed to classify the abundance of bacteria to the taxonomic level of species or genera in metagenomic data sets. Within metagenomic data sets there are often hundreds to thousands of individual reads from the more abundant bacterial species, potentially providing enough evidence for subtyping (*Sahl et al., 2015*). There have been a number of recent publications describing new software dedicated to taxonomic analysis of shotgun metagenomic data, reflecting the intense interest in this field (*Lindgreen, Adair & Gardner, 2015*). Several methods use unsupervised clustering of sequence reads belonging to the same species based on patterns of coverage covariance (*Sharon et al., 2013*; *Nielsen et al., 2014*; *Alneberg et al., 2014*). These methods require processing of multiple metagenomic data sets simultaneously and are not explicitly designed for subtyping, although they could potentially be adapted for that purpose. Reference-based strategies attempt to assign individual reads to species based on existing completed or draft genome sequences, using alignment based methods or k-mer counting (*Naccache et al., 2014*; *Huson et al., 2007*; *Schaeffer et al., 2015*; *Darling et al., 2014*). Some supervised tools have been developed specifically to type bacterial "strains" (usually taken to mean a taxonomic rank below the species level) (*Sahl et al., 2015*; *Cleary et al., 2015*; *Zagordi et al., 2011*; *Luo et al., 2015*; *Hong et al., 2014*; *Ahn, Chai & Pan, 2015*). Here we report the development of a new metagenome based subtyping aimed-specifically at the well-studied bacterial pathogen, *S. aureus*. Unlike other strategies, we focused on a single important species with in the same metagenome, making use of the extensive set of reference genome sequences now available to create a SNP matrix corresponding to 40 major subtypes. The output of the analysis was the estimate of proportions of subtype-specific sequences.

*S. aureus* is one of the most common hospital infections, often causing diseases with poor outcome (*King et al., 2006*; *Dulon et al., 2011*). The bacterium is also a problem outside the hospital as a community-acquired infection in humans (*Klevens et al., 2007*; *Kuehnert et al., 2006*), livestock (*Rinsky et al., 2013*; *Price et al., 2012*) and other animals (*Paterson et al., 2015*). *S. aureus* is a common asymptomatic colonizer of humans. It is suggested that 20–50% of human anterior nares (noses) are persistently colonized with *S. aureus*, with 60–100% of individuals harboring *S. aureus* at some point in their lifespan (*Van Belkum et al., 2009*; *Lamers et al., 2011*; *Liu et al., 2015*). The population of *S. aureus* asymptomatically colonizing the nose in healthy individuals is thought to be a major source for transmission (*Von Eiff et al., 2001*).

*S. aureus* strains can be classified into a limited number of clonal lineages of related MLST sequence types (clonal complexes (*Feil et al., 2004*)) , which differ in their geographical distribution and propensity to cause human diseases. MRSA strains containing the SCCmec cassette, are more common in some clonal lineages than others (28), as is the acquisition of *vanA* genes to produce VRSA (vancomycin resistant *S. aureus*) (*Kobayashi, Musser & DeLeo, 2012*). It is important to understand the genetic diversity (population structure) of *S. aureus* strains that colonize the different body sites in order to understand how commensal strains present in healthy human population might act as a predisposing factor for future invasive infections. Using our subtyping scheme based on metagenomic data, we performed epidemiological modeling to understand whether there is any association with the demographic and life history characteristics collected using the responses that subjects gave to an extensive survey, and the different strains of *S. aureus* identified at each body site.

## MATERIALS AND METHODS

### Classifying *S. aureus* subtypes based on a binomial mixture model

We used *binstrain* software (*Joseph et al., 2014*), implemented in the R language (*R Core Team, 2014*) to perform *S. aureus* subtype classification. *binstrain* uses a binomial mixture model to estimate the proportion of subtypes based on a DNA alignment against a reference (SA_ASR, described below) and a matrix of SNPs that distinguish different genetic subtypes (construction of the matrix described below). *binstrain* assumes a binomial probability distribution, $p_i$ of observing a SNP, $x_i$ in the entire genome and $n_i$ denotes the total nucleotide coverage at position $i$. $Z_{i,j}$ is an indicator function specifying whether $j^{\text{th}}$ strain has a SNP at $i^{\text{th}}$ position. In the final version of the classifier, we used 102,057 SNP positions across the genome to classify *S. aureus* into 40 subtypes.

$$x_i \sim \text{Binom}(n_i, p_i), \quad i = 1, \ldots 102{,}057$$
$$p_i = \beta_i Z_{i,1} + \beta_2 Z_{i,2} + \cdots + \beta_{40} Z_{i,40}, \quad i = 1, \ldots 102{,}057.$$

The estimation of $\beta_i$ indicates the proportion of *S. aureus* reference strain-specific SNPs present in a clinical or purified sample. At the strain–specific SNP positions, there will be only a few $\beta_i$'s that affects $p_i$. Other $\beta_i$'s have no impact on $p_i$ because their corresponding $Z_{ij}$ are 0's, which makes it a sparse design matrix. We utilized this sparsity of the design matrix in order to perform a well-established procedure to estimate all the $\beta_i$'s using quadratic programming (*Joseph et al., 2014*).

### *S. aureus* ancestral sequence regeneration

In order to elucidate the population structure of *S. aureus* and to select distinct subtypes, we performed ChromoPainter (http://www.paintmychromosomes.com) and fineSTRUCTURE (*Yahara et al., 2013*; *Lawson et al., 2012*) analysis on 43 completed genomes downloaded from NCBI (Table S1). The ChromoPainter analysis was applied using the linkage model to the genome-wide haplotype data generated from a whole genome progressive MAUVE (*Darling, Mau & Perna, 2010*) alignment of the 43 genomes to generate a coancestry matrix, which was used by the fineSTRUCTURE algorithm to perform

model-based clustering using the Bayesian MCMC approach. This analysis indicated the presence of 19 distinct populations of *S. aureus* out of the 43 completed genomes used (Table S1). We selected one genome sequence from each of the 19 groups (choosing randomly in the cases where more than one genome sequence was represented) and reconstructed the phylogenetic tree using the maximum likelihood approach by RAxML (*Stamatakis et al., 2012*) with 100 bootstraps based on our MAUVE alignment. The ancestral sequence of *S. aureus* (SA_ASR) (2,872,915 bp) (Data S1) was generated using the baseml program in the PAML package (*Yang, 2007*) with the whole genome alignment and phylogenetic tree as inputs. We implemented GTR nucleotide substitution model, with 5 gamma rate categories, assuming that the model was homogeneous across all the sites.

## Development of the *S. aureus binstrain* v2 SNP matrix

We initially constructed a *binstrain* (*Joseph et al., 2014*) SNP matrix (v1) based on the 19 reference genomes of the SA_ASR. We tested this prototype classifier against 2,692 diverse *S. aureus* genomes downloaded as FASTQ files from NCBI. We first mapped the reads from each genome against the SA_ASR to generate the base call and coverage (average read depth) in each position in the mpileup output format. The alignment was performed using the Burrows-Wheeler Aligner (BWA) (Version: 0.6.1-r104) short-read aligner (*Li & Durbin, 2009*) by specifying the maximum number of gap extensions (e) to be 10. The resultant short-read alignment files for each samples were converted to mpileup format using the mpileup option in SamTools software along with the –B option that disables probabilistic realignment for the computation of base alignment quality (BAQ). When challenged using *binstrain* we found that only 1,379 of the 2,692 (51.22%) genomes estimated a beta value $\geq 0.8$. We constructed a second version of the SNP pattern file based on a whole genome progressiveMAUVE alignment with an additional 21 draft *S. aureus* genome chosen to cover the estimated 40 populations within this set (Table S2). We filtered out all the SNP positions where any of the *S. aureus* strains and two genomes of the near-neighbor *S. epidermidis* species (*Méric et al., 2015*) (*S. epidermidis ATCC12228* (accession number: NC_004461.1) and *S. epidermidis RP62A* (accession number: NC_002976.3) had a SNP at the same position. The resultant v2 SNP pattern file contained a total of 102,057 SNP positions across the entire core genome. The SNP pattern file contained SNPs unique for each reference strain as well as SNPs shared among more than one reference strains represented in this study. To determine specificity and sensitivity of the v2 classifier we tested 2,114 diverse *S. aureus* strains from the NCBI SRA database with ST determined by SRST software (*Inouye et al., 2012*), downsampled to genome coverage between 0.025X and 0.75X (Data S2).

## Testing subtype classification using simulated sequence data

In order to test the *S. aureus* subtype classification scheme we used ART software (*Huang et al., 2012*) to simulate single end 100 bp Illumina read sequence data based on assembled *S. aureus* genome templates. ART simulates sequencing reads by mimicking the output of the Illumina sequencing process with empirical error models summarized from large recalibrated sequencing data. FASTQ files were generated based on 40 *S. aureus*
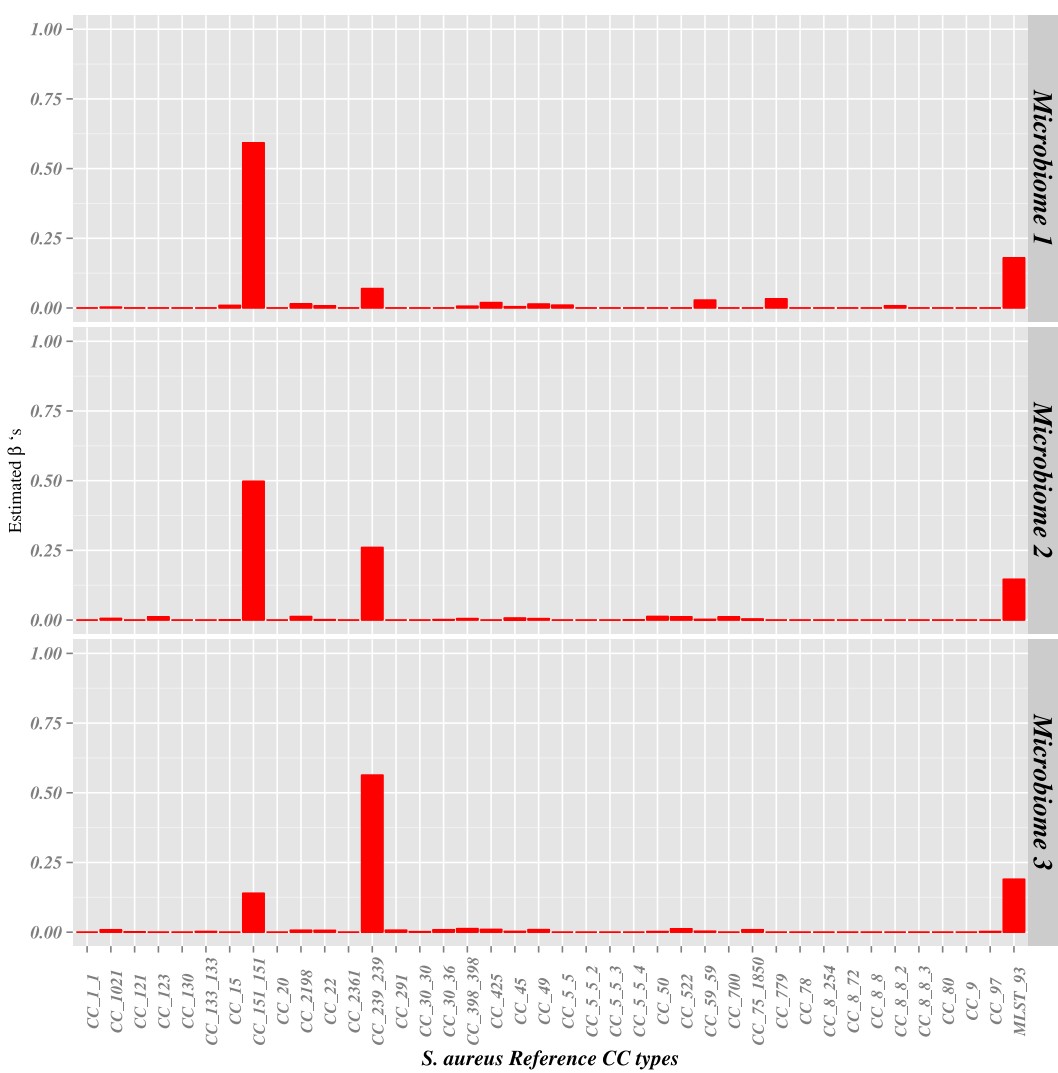

**Figure 1** **Classification of from synthetic microbiomes.** *S. aureus* reference genomes (CC_151, CC_239 and MLST_93) were added to *Propionibacterium acnes* (200X), *S. epidermidis* (100X), *Streptococcus mitis* (7X), *Bacteroides vulgatus* (5X), *Staphylococcus haemolyticus* (4X) and *Staphylococcus saprophyticus* (3X). (1) **microbiome 1**–CC_151_151 (0.5X), CC_239_239 (0.1X) and MLST_93 (0.25X) (2) **microbiome 2**— CC_151_151 (1.5X), CC_239_239 (1X) and MLST_93 (0.75X) and (3) **microbiome 3** –CC_151_151 (1.0X), CC_239_239 (5.0X) and MLST_93 (2.5X). Barplots show the results from *binstrain* with the v2 matrix. The 'other' category represent diverse subtypes called with low probability.

reference genomes used in this study at varying coverages. FASTQ files were simulated for each strain 9 times separately at the following genome coverages (X): 0.000625X, 0.0125X, 0.025X, 0.05X, 0.1X, 0.5X, 0.75X, 1X, 2.5X and 5X. In addition, we simulated 3 artificial microbiome samples by individually simulating sequencing reads of the major bacteria present in a healthy anterior nares identified by the HMP (*Human Microbiome Project Consortium, 2012a*) and concatenated them all together with 3 random *S. aureus* reference genomes (CC_151, CC_239 and MLST_93) (Fig. 1). The bacteria simulated were *Propionibacterium acnes* (200X), *S. epidermidis* (100X), *Streptococcus mitis* (7X), *Bacteroides vulgatus* (5X), *Staphylococcus haemolyticus* (4X) and *Staphylococcus saprophyticus* (3X).The

proportions of the 3 *S. aureus* genomes in each of the 3 simulated microbiome samples were: (1) **microbiome 1**—CC_151_151 (0.5X), CC_239_239 (0.1X) and MLST_93 (0.25X) (2) **microbiome 2**—CC_151_151 (1.5X), CC_239_239 (1X) and MLST_93 (0.75X) and (3) **microbiome 3**—CC_151_151 (1.0X), CC_239_239 (5.0X) and MLST_93 (2.5X).

## Whole genome neighbor-joining phylogeny

2,114 randomly chosen *S. aureus* strains were downloaded from the NCBI and converted to FASTQ format. The files were downsampled to 50× coverage using a custom script based on the seqtk tool (see https://github.com/lh3/seqtk and https://gist.github.com/rpetit3/9c623454758c9885bf81d269e3453b76) and mapped against the *S. aureus* N315 genome using BWA (*Li & Durbin, 2009*). SNPs and indels identified in the core genome using GATK (56) were exported as VCF files and uploaded into a custom in-house Postgres database called 'Staphopia' (R Petit et al., 2016, unpublished data). We identified all the SNP positions and extracted the base calls at each site. We then subtracted from the matrix all SNP positions that overlapped any indel called in any of the genomes. The concatenated bases were used to create an aligned multi-fasta file of 339,823 bp (https://figshare.com/articles/aligned_subtyping_strains_fasta/3581598). A distance matrix and neighbor joining tree were calculated using the R *phangorn* package (*Schliep, 2010*). *S. aureus* MLST genotypes were assigned using the SRST2 (*Inouye et al., 2014*) program running on the raw FASTQ files.

## Classification of NCBI genome groups

DNA fasta files of representative genomes of 25 *S. aureus* genome groups were downloaded in August 2015 (http://www.ncbi.nlm.nih.gov/genome/?term=Staphylococcus+aureus). Synthetic 5X coverage was generated for each using ART(35). MLST types were ascertained using SRST2(58) and *binstrain* subtypes were ascertained using the v2 matrix.

## Sequence data analysis and statistical modeling for SNP based genotyping of *S. aureus* strains in the HMP and NYC subway metagenome projects

For this study we obtained raw mwgs sequence data in FASTQ files for 1,265 samples from the HMP ftp site (ftp://public-ftp.hmpdacc.org/Illumina). The HMP carried out 2 phases of mwgs, performed using the Illumina GAIIx platform with 101 bp paired-end reads. For Phase 1, 764 samples were chosen from 103 adults and for Phase II, 400 samples were chosen from 67 adults. Samples were chosen covering 16 body sites. The Phase 1 data sets have been described previously (*Human Microbiome Project Consortium, 2012a*; *Human Microbiome Project Consortium, 2012b*). We also used mwgs data from DNA samples extracted from the New York subway (*Afshinnekoo et al., 2015*). The 125 × 125 bp paired end sequence reads were prepared using an Illumina HiSeq 2500 instrument. Sequence data was downloaded from the NCBI SRA repository (project PRJNA271013).

   We mapped the FASTQ files from these projects at against the SA_ASR to obtain mpileup format data using BWA as described above and called the subtypes present using *binstrain*. From the mpileup output we calculated the average gene coverage mapping to the *S. aureus* core genome. In addition to the core genome, we also determined the number

of reads mapping to a representative *S. aureus* *mecA* gene using BWA (*Li & Durbin, 2009*) with same parameters mentioned above (Locus AKR51832.1 gi:899756207).

## Selection of metadata from the HMP and epidemiological modeling

A large amount of demographic and clinical data was collected for each of the individuals sampled for the HMP. We obtained access to the most recent version of these data through a formal request to the NCBI dbGap database (accession phs000228.v3.p1). We used the metadata from the 170 healthy individuals (phase I and II) from whom the HMP mwgs sequence data were generated. Because of the generally high level of health of the subjects, for most clinical variables there were too few cases to have realistic odds of association. The binary categorical variables (exposure variables) from the metadata, which we investigated in relation to presence of *S. aureus* and/or a particular *S. aureus* subtype in a body site were gender, breastfed or not, tobacco use, insurance information and history of previous surgery (Table S3). Other categorical variables used were diet (Meat/fish/poultry at least three days per week, Meat/fish/poultry at least one day but not more than two days per week and Eggs/cheese/other dairy products, but no meat/fish/poultry), race/ethnicity (Hispanic, Asian, non-Hispanic black and non-Hispanic white) and BMI (<22, 22–25 & >25). Age was treated as a continuous variable that ranged from 18 to 40 years of age (Table S3). For the epidemiological analysis, we included only those body sites where *S. aureus* was detected in at least 20% of the samples. We also grouped the body sites into three superclasses: airways (anterior nares); oral cavity (attached keratinized gingiva, buccal mucosa, palatine tonsils, saliva, supragingival plaque and tongue dorsum) and skin (right and left retroauricular crease). There were a total of 840 samples collected from 133 participants in the HMP, used in the epidemiological analysis (described below).

We performed both binary and multinomial (with 4 outcomes) logistic regression to identify predictors for *S. aureus* detection among HMP participants. The binary outcome indicated whether the presence of *S. aureus* was detected or not detected (reference), while the 4 outcomes for the multinomial logit model were the presence/detection of *S. aureus* CC8, CC30, any other *S. aureus* CC types and no detection of *S. aureus* (reference). Odds ratios were estimated by fitting generalized linear mixed models using SAS PROC GLIMMIX (Version 9.4, Cary, NC) with main site and other exposure variables (described above) as fixed effects and random effects for subject in order to assess any possible association of the exposure variables and the presence/detection of *S. aureus*.

## Association of *S. aureus* HMP subtypes with body site and individual subjects

For each HMP project that contained *S. aureus* reads we prepared a Hamming distance matrix between samples based on the shared presence of subtypes inferred from the *binstrain* beta estimates. We only used subtypes with at least a beta value of 0.2, in order to mitigate the possible effect of overcalling subtypes. From this distance matrix we performed a principal components analysis and mapped the distribution of samples by body site and individual subject. In order to test whether individual body sites were more likely to contain similar subtypes we calculated the total Hamming distance between all samples of the same

body type and compared the value to 10,000 random draws of the same number of samples from the total set of *S. aureus* positive samples.

### Spatial relationships in NYC subway sampling sites

We used the RgoogleMaps package (*Loecher & Ropkins, 2015*) to plot the subway stations reporting *S. aureus* subtypes based on geographic coordinates from supplemental data in *Afshinnekoo et al. (2015)*. We used these data to calculate a matrix of straight line distances between all 466 subway stations and then performed a regression between geographic distance in km and the Hamming distance of *S. aureus* subtype composition. Finally, we looked for clustering between *S. aureus* subtypes by comparing the geographic distance between stations reporting a subtype and 1,000 permutations of *n* random stations in NYC (where *n* was the number of stations reporting the subtype).

## RESULTS

### A *S. aureus* subtyping scheme for metagenomic data

There are two main challenges in subtyping a bacterial species based on metagenomic data: missing loci and mixed strain composition. The first problem arises when coverage is too low to guarantee the presence of a read containing a given sequence (SNP, indel, k-mer etc.) in the target genome. The missing loci problem obviates against using schema that have an absolute requirement for a particular sequence being represented, such as VNTR and MLST. The mixed strain issue means that the scheme cannot assume that only one subtype is present. For these reasons we chose to use a software, *binstrain* (*Joseph et al., 2014*) that we had previously developed for distinguishing mixed clonal populations of *Chlamydia trachomatis*.

*binstrain* used a binomial mixture model to separate the relative abundance of subtype-specific SNPs (other types of variants can be used but in this work we used only SNPs) found across the population of sequence reads. The software, implemented in R, integrated the results from SNP positions found across the target genome but was not dependent on all loci being present in the sample. Further, since *binstrain* was developed to determine sample mixtures, results were expressed as the proportion of each subtype present in the sample. *binstrain* required as input a matrix of SNPs assigned to each subtype. To generate the matrix we defined subtypes based on currently available *S. aureus* genome sequence data. We first generated an inferred ancestral reference genome sequence (SA_ASR) using baseml (see 'Materials and Methods') based on the MAUVE alignment of 19 *S. aureus* genomes from different clonal complexes (Data S1). The SA_ASR was used it as a reference genome for alignment of sequence reads to reduce possible recruitment biases that could occur between a modern reference and more or less closely related strains. We then developed a SNP matrix based on a training set of 2,692 genetically diverse *S. aureus* strains downloaded from the Sequence Read Archive database. Based on the results of clustering of the core portion of chromosomes these genomes fell into 40 distinct populations. We picked 40 genome sequences to represent these populations. For 19 of the populations we determined that there was a representative *S. aureus* completed genome (from those used to make the SA_ASR; Data S1). For the 21 remaining populations, we randomly picked one genome as

representative. We created a whole genome alignment of the core portion of the 40 genomes using MAUVE and created a matrix (v2) of each SNP position relative to the inferred ancestral nucleotide in SA_ASR. We named the subtypes to match as near as possible the established MLST-based clonal complex (CC) (33) designation. The forty representative genomes contained a total of 102,057 SNP positions in 2,872,915 bps of the *S. aureus* chromosome. In the v2 scheme, the clonal complexes CC8, CC5 and CC30 were represented by multiple reference strains. In order to make the analysis compatible with the existing CC nomenclature, we collapsed CC8, CC5 and CC30 into one subtype each for visualizations. Versions of figures made using the uncollapsed values are available on the github site for the manuscript (https://github.com/Read-Lab-Confederation/staph_metagenome_subtypes).

Before attempting to subtype *S. aureus* data from the HMP metagenomic project using the v2 matrix, we ran experiments on simulated data (100 bp Illumina reads generated by the ART program (*Huang et al., 2012*)) to test the sensitivity and specificity of the method. First, we showed that the presence of up to 320X coverage of sequence reads from the near-neighbor species *S. epidermidis (S. epidermidis ATCC12228* (accession number: NC_004461.1) and *S. epidermidis RP62A* (accession number: NC_002976.3) did not affect the accuracy of the *binstrain* assignment (see 'Materials and Methods'). This was an expected result as SNPs in the *S. epidermidis* that overlapped those used by *binstrain* for subtype identification had been removed from the matrix (see 'Materials and Methods') and therefore is effectively 'invisible' to this typing method. We next created 3 artificial nasal metagenomes consisting of reads from common nasal microflora (*Human Microbiome Project Consortium, 2012a*), and included synthetic reads from 3 randomly chosen *S. aureus* genomes that had a combined length of ∼2.9 Mb (i.e., ∼1× genome coverage). Despite being less than 0.5% of the total reads in the artificial metagenome, we accurately determined the approximate relative proportions of the *S. aureus* subtypes (Fig. 1). In each test, 9–14% of the variation was attributed to a range of subtypes other than the three used in creation of the synthetic microbiomes (marked as "other" in Fig. 1). We believe these miscalls come from a combination of sequence error introduced by the ART software and *binstrain* assignment of probably to multiple subtypes to SNPs in internal branches of the species phylogeny. For this reason we chose a $\beta$ value > 0.2 cutoff as a conservative lower bound threshold for calling the presence of a subtype being present as at least a minor component in situations where more than one *S. aureus* CC type might be present in the metagenomic sample based on the results from *binstrain*. A $\beta$ value threshold of >0.8 was chosen as a threshold to indicate the sample was dominated by one subtype.

To further assess the sensitivity and specificity of the *binstrain* test, we performed analysis on 10 simulated read sets of between 0.00625X and 5X genome coverage for each of the 40 subtype-defining genomes and repeated the analysis nine times. The *binstrain* test showed a low false positive rate with the simulated data: we only found strains assigned to the wrong subtype on 748/3600 times (21%), and this only occurred when the coverage was low (<0.0125X coverage) and the misclassification was to between subtypes that both mapped to CC5 (Fig. 2; Fig. S1). The false negative rate was high at the lowest coverages, but at 0.5X coverage > 95% of strains were correctly assigned.

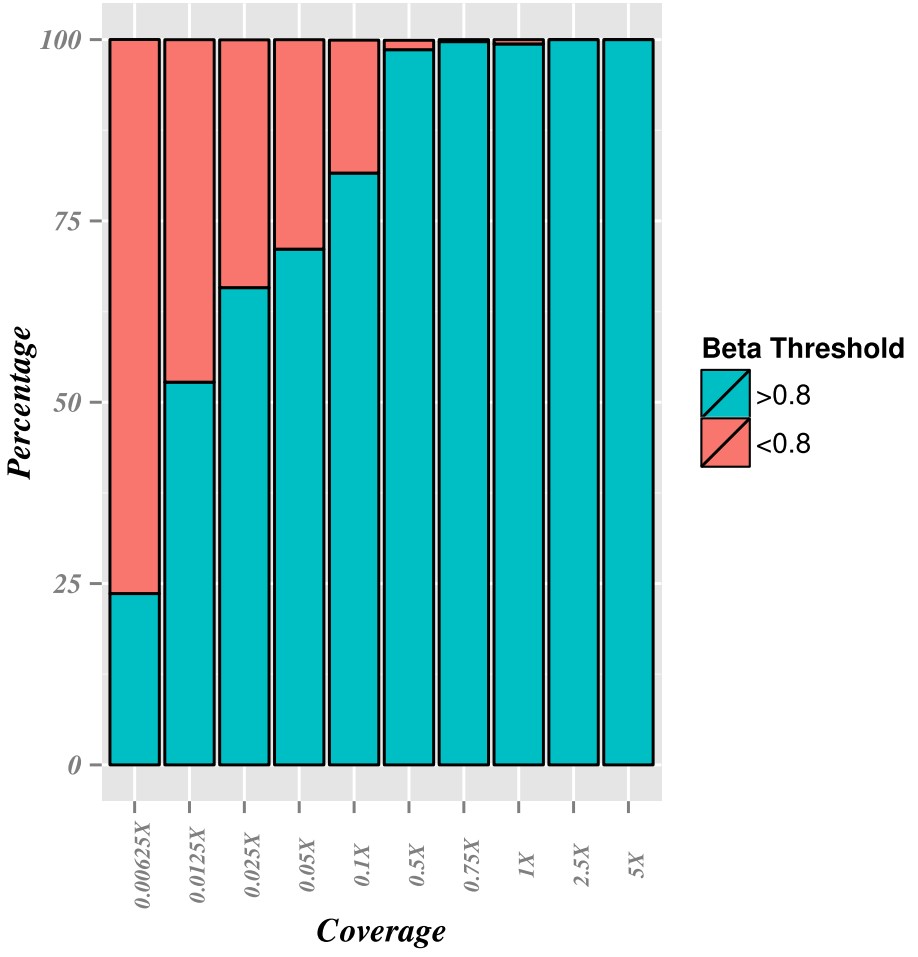

**Figure 2  Sensitivity of *binstrain* assignment at different coverages.** A total of 3,600 FASTQ files were simulated for each of 40 *S. aureus* strains chosen for genetic diversity 9 times separately at the following genome coverages (X): 0.000625X, 0.0125X, 0.025X, 0.05X, 0.1X, 0.5X, 0.75X, 1X, 2.5X and 5X, and were classified using *binstrain* with the v2 matrix. The sensitivity of *binstrain* assignment to the correct *S. aureus* CC type are shown as percentages where each of the 40 CC types within the specified simulated genome coverage was correctly called above a beta (b) >0.8 score. The green color indicates the percentage of situations where the correct CCtype was called above a $\beta$ of 0.8, while the orange color indicates situations where the correct CCtype/strain was called below a $\beta$ score of 0.8.

As a further test, we ran the v2 SNP matrix against a further 2,114 *S. aureus* FASTQ files downloaded from the NCBI SRA, representing a random selection of genotypes. Each strain was assigned to a MLST sequence type (ST) based on the results of the SRST software (34). Each FASTQ file was randomly downsampled to 0.025X–0.75X coverage and run through the *binstrain* classifier (Tables 1A, 1B and S2). Compared to the synthetic FASTQ files reported above we found that the sensitivity of the test was higher with data sets from real sequencing projects (∼84% on 0.025X coverage data even with a restrictive $\beta$ cutoff of 0.9 or above (Table 1A)). When the positive calls were mapped onto a whole genome phylogeny of the *S. aureus* isolates, we confirmed that there were no major lineages systematically excluded (Fig. 3). We measured specificity using the logic that strains with the same ST should be placed in the same subtype; thus we counted the largest subtype for each ST as

Table 1 **Summary of the sensitivity and specificity of v2 subtyping matrix for *S. aureus*.** (A) Sensitivity of the v2 subtyping matrix on 2188 *S. aureus* FASTQ files from the NCBI SRA at different cutoff score. (B) Specificity of the v2 subtyping matrix on 2188 *S. aureus* FASTQ files from the NCBI SRA at different cut-off $\beta$ scores.

| Coverage | $\beta \geq 0.90$ | $\beta \geq 0.80$ | $\beta \geq 0.70$ | $\beta \geq 0.60$ |
|---|---|---|---|---|
| **(A)** | | | | |
| 0.025X | 83.91% | 86.19% | 86.80% | 87.41% |
| 0.05X | 87.04% | 89.03% | 89.64% | 89.97% |
| 0.1X | 88.70% | 91.77% | 92.33% | 92.76% |
| 0.5X | 85.71% | 93.18% | 95.93% | 97.20% |
| 0.75X | 84.34% | 92.38% | 95.78% | 97.25% |
| **(B)** | | | | |
| 0.025X | 91.69% | 93.94%% | 94.54% | 94.98% |
| 0.05X | 94.70% | 96.45% | 97.05% | 97.21% |
| 0.1X | 94.37% | 97.17% | 97.46% | 97.46% |
| 0.5X | 90.65% | 95.53% | 96.07% | 96.46% |
| 0.75X | 90.26% | 96.69% | 97.53%% | 98.25% |

'true' (Table 1B). Based on these criteria we achieved >91% specificity even at the lowest coverage and most stringent $\beta$ cutoff. We noted that strains classified as ST5 had a lower specificity (Table S4). Based on these tests, we decided that a coverage of 0.025X *S. aureus* genome equivalents was a reasonable threshold for our HMP metagenomic data, giving us a large sample set with an acceptable specificity and sensitivity of ∼89% and 92%, respectively.

As a final test, we downloaded 25 genomes representative of the major *S. aureus* groups defined by NCBI (http://www.ncbi.nlm.nih.gov/genome/?term=Staphylococcus+aureus). *binstrain* ($\beta$ cutoff > 0.65) matched 15/25 of the genomes to the subtype matching its ST ascertained by SRST2 (Table S5). These 15 groups represented >95% of the 6,882 genomes grouped by NCBI, in the line with the specificity achieved with the genome projects downloaded from SRA (paragraph above). The 10 groups not matched represented rare *S. aureus* subtypes.

## Assignment of *S. aureus* subtypes in the HMP metagenomic dataset using whole genome subtyping

Having developed and tested the v2 *S. aureus binstrain* classifier we attempted to call the subtypes present in 1,263 whole metagenomic sequencing samples from the healthy human cohort of the phase 1 and 2 of the HMP (300 subjects). We found at least one sequencing read mapping to the SA_ASR sequence in 348 of the samples (27.5%) isolated from 110 (36.3%) of the subjects (Fig. S2). The presence of the species was variable across body sites, most commonly found in the left and right retroauricular creases and anterior nares (100%, 90% and 57%, respectively) and least common in the stool and subgingival plaque (6% and 5%, respectively) (Table 2). While the presence of the *S. aureus* reads in a sample will be dependent on factors such as the complexity of the microbiome and the amount of sequence data collected, this result was in line with estimates of *S. aureus* presence based on bacterial culture (*Van Belkum et al., 2009*).

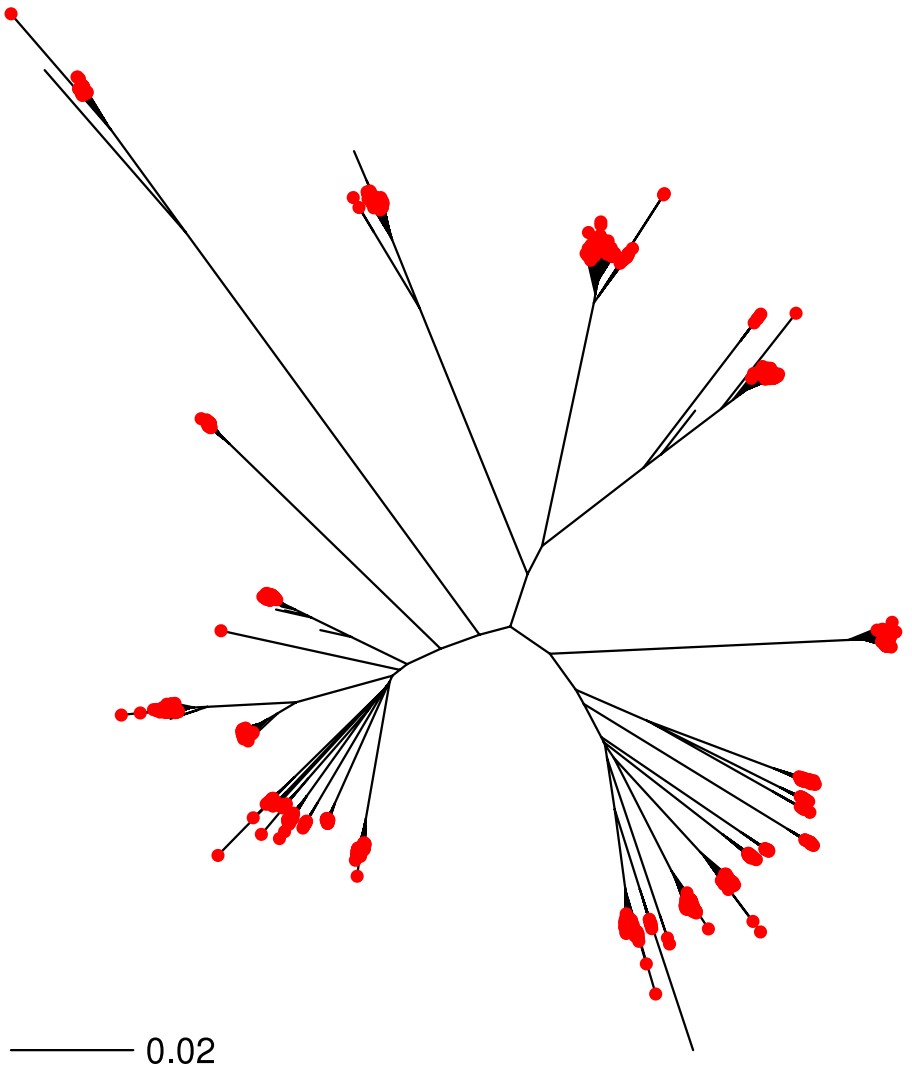

0.02

**Figure 3** **Phylogenetic distribution of subtyped *S. aureus* strains.** 2,188 randomly chosen *S. aureus* shotgun genome projects were downsampled to create a new FASTQ file with 0.75X coverage. Each FASTQ file was classified using *binstrain* with the v2 matrix as a single subtype ($\beta > 0.8$ cutoff) is marked with a red point on the *S. aureus* phylogeny. The neighbor-joining tree of the 2,188 strains was produced from a distance matrix derived from concatenation of all SNP positions mapped against reference strain *S. aureus* N315. The figure illustrates that the v2 matrix has minimal phylogenetic bias in classification of strains.

Of the 348 *S. aureus* positive samples, 321 had a *S. aureus* core coverage > 0.025X (Table 2). *S. aureus* was more prevalent at this level of coverage in the anterior nares, retroauricular creases and tongue dorsum. 165 (51%) of these samples were dominated by one subtype (defined at containing a subtype with $\beta$ value > 0.8). The most commonly detected subtypes were CC30, CC8 , CC45, CC398 and CC5 (present in 112 (35%), 72 (22%), 32 (10%), 29 (9%) and 26 (8%) samples, respectively) (Fig. 4). There were only 2 subtypes not detected in any of the samples: CC123 and CC49. The US origin of the samples was reflected by the common occurrence of ST8 (representing lineages such as USA300 and

**Table 2** *S. aureus* **positive HMP body sites based on having at least one read mapping to the SA_ASR sequence.** Percentages based on total number of samples for that body site.

| Body site | Total number of samples | Number of *S. aureus* positive samples | Number samples with >0.025× *S. aureus* coverage |
|---|---|---|---|
| Anterior nares | 137 | 78(57%) | 68(50%) |
| Attached keratinized Gingiva | 14 | 4(29%) | 4(29%) |
| Buccal mucosa | 185 | 56(31%) | 56(30%) |
| Hard Palate | 1 | 1(100%) | 1(100%) |
| Left retroauricular crease | 23 | 23(100%) | 23(100%) |
| Palatine tonsils | 19 | 7(37%) | 6(32%) |
| Posterior fornix | 108 | 11(10.20%) | 9(8.33%) |
| Right Antecubital fossa | 1 | 1(100%) | 1(100%) |
| Right retroauricular crease | 31 | 28(90%) | 28(90%) |
| Saliva | 7 | 2(28.57%) | 1(14%) |
| Stool | 251 | 14(6%) | 7(0.3%) |
| Subgingival plaque | 19 | 1(5%) | 1(5%) |
| Supragingival plaque | 210 | 65(31%) | 37(18%) |
| Tongue dorsum | 221 | 82(37%) | 82(37%) |

USA500). CC30 and CC45 are common colonization isolates worldwide, whereas CC239 and CC22; prevalent in hospitals in Europe, Asia and Latin America (*Baines et al., 2015*; *Knight et al., 2012*; *Stanczak-Mrozek et al., 2015*), were infrequent in this study (present in 1 and 2 samples, respectively).

The median genome coverages of samples from the anterior nares, tongue dorsum and retroauricular creases were higher than that of the other body sites (with the exception of the hard palate where $n = 1$; Fig. S3). However, in all but 21 out of 321 samples (93%) *S. aureus* genome coverage was below 2×. A sample from the anterior nares (NCBI SRA accession SRS011105) was a striking exception, with $> 70×$ coverage of a CC59 S. aureus genome. Although the subtype schema was based on the core genome, we also calculated the coverage of the accessory *mecA* gene, necessary for MRSA. 55/1263 (4%) samples had reads mapping to *mecA*. There was no consistent relationship between the coverage level of the *S. aureus* genome and the coverage of *mecA*, even when we looked at the adjusted coverage (*binstrain β* multiplied by total coverage) for individual subtypes. This is likely because *S. aureus* subtypes can consist of both MRSA and non-MRSA strains (MSSA) (*Enright et al., 2002*; *Nübel et al., 2008*) and *mecA* genes occur in *S. epidermidis* and other staphylococci (*Diekema et al., 2001*) that may also be present in the sample. There were no *mecA* reads present in accession SRS011105, indicating that the CC59 strain present at 70× coverage was MSSA.

### *S. aureus* subtype distribution across subjects and body sites

We found differences in the distribution of *S. aureus* subtypes between across body sites in the HMP data set (Figs. 5A–5C). For example, CC30 dominated the tongue dorsum, whereas CC8 was most prevalent in the buccal mucosa. The most striking bias in distribution was seen in the livestock associated CC398 subtype, which was found in the tongue dorsum in

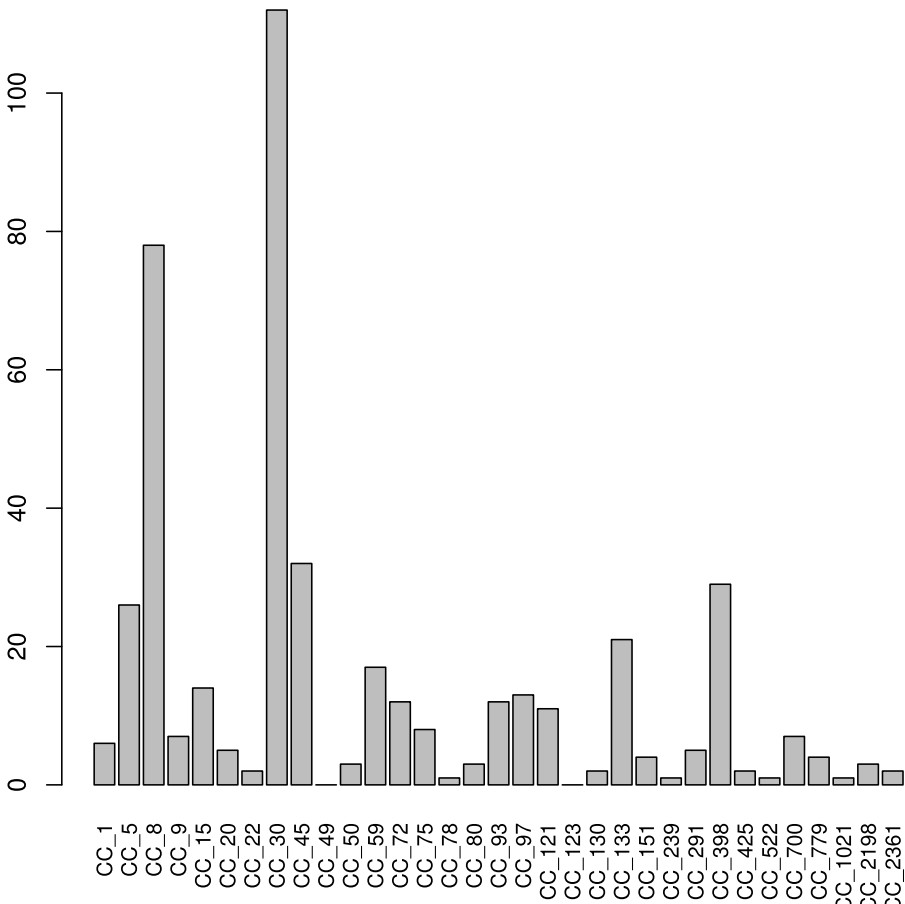

**Figure 4** *S. aureus* **subtypes in HMP samples.** 321 samples from the HMP project with a *S. aureus* core coverage > 0.025X project were classified using *binstrain* with the v2 matrix. The figure shows counts of the number of samples with each subtype present with beta > 0.2.

26/29 (90%) samples (Fig. 5C). We found that all the positive body site samples contained, in addition to a small number of rare SNPs, a synonymous SNP within the RNA polymerase gene that is conserved in almost all CC398 strains (Fig. 6) but rarely found outside this subtype.

Each sample was scored with a value of 1 for every subtype with a $\beta$ value > 0.2 and then a distance matrix based on Hamming scores between samples was created. There was a significant association between Hamming score distance and body site (PERMANOVA (*Anderson, 2001*), $p < 0.001$, $df = 8$) but the model explained only ~11% of the variance. However, the result was tempered by finding a significant difference in beta dispersion between body sites ($p < 0.001$), which might have caused spurious association. As an alternative approach, we compared the sum of Hamming distances between samples of the same body type to 10,000 randomly selected samples (Table 3). The posterior fornix, buccal mucosa, stool and supragingival plaque had total Hamming scores that were <5% of those found in random sampling.

Two body sites on the same individual may be more similar to each other than the same sites in other individuals because of intrapersonal spread of bacteria. There were

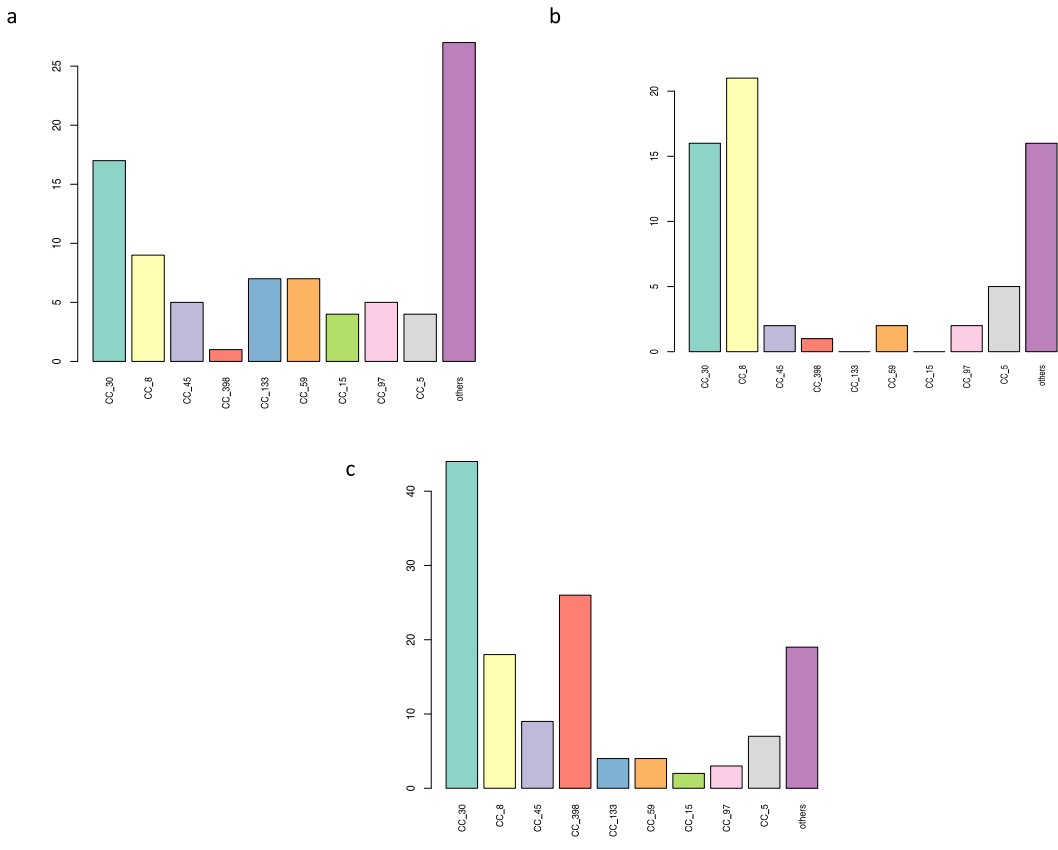

**Figure 5** **Major *S. aureus* subtypes at three HMP body sites.** The data from Fig. 4. Broken down by three body sites, showing the most most subtypes. (A) anterior nares, (B) buccal mucosa, (C) tongue dorsum.

108 individuals within the HMP group with at least one body site containing >0.025X *S. aureus* genome coverage. There was a significant association between subject and *S. aureus* composition (PERMANOVA, $p = 0.005$, $df = 107$, $R^2 = 0.4$). Unlike the body site categories tested above, there were no significant differences in dispersion of Hamming distances between individuals. An interaction model for body site and subject accounted for ~47% of variance and had a p value of 0.001. As an additional confirmation of this result, we calculated the total Hamming distance of all *S. aureus* communities within the same individual (including both Phase I and Phase II data) and found it was lower than 10,000 random replicates of the same number of samples.

### Biotic and abiotic factors associated with *S. aureus* and its subtypes

We performed epidemiologic modeling using generalized linear mixed models to assess whether any metadata variables on the subjects of the study collected by the HMP were associated either with the presence of *S. aureus,* or with a specific subtype. In order to increase power, we aggregated body sites into three categories: airways (anterior nares), oral cavity (attached keratinized gingiva, buccal mucosa, palatine tonsils, saliva, supragingival plaque and tongue dorsum) and skin (right and left retroauricular crease). In the binary outcome logistic regression model, at an alpha level of 0.1, main body site (*p*-value < 0.001),
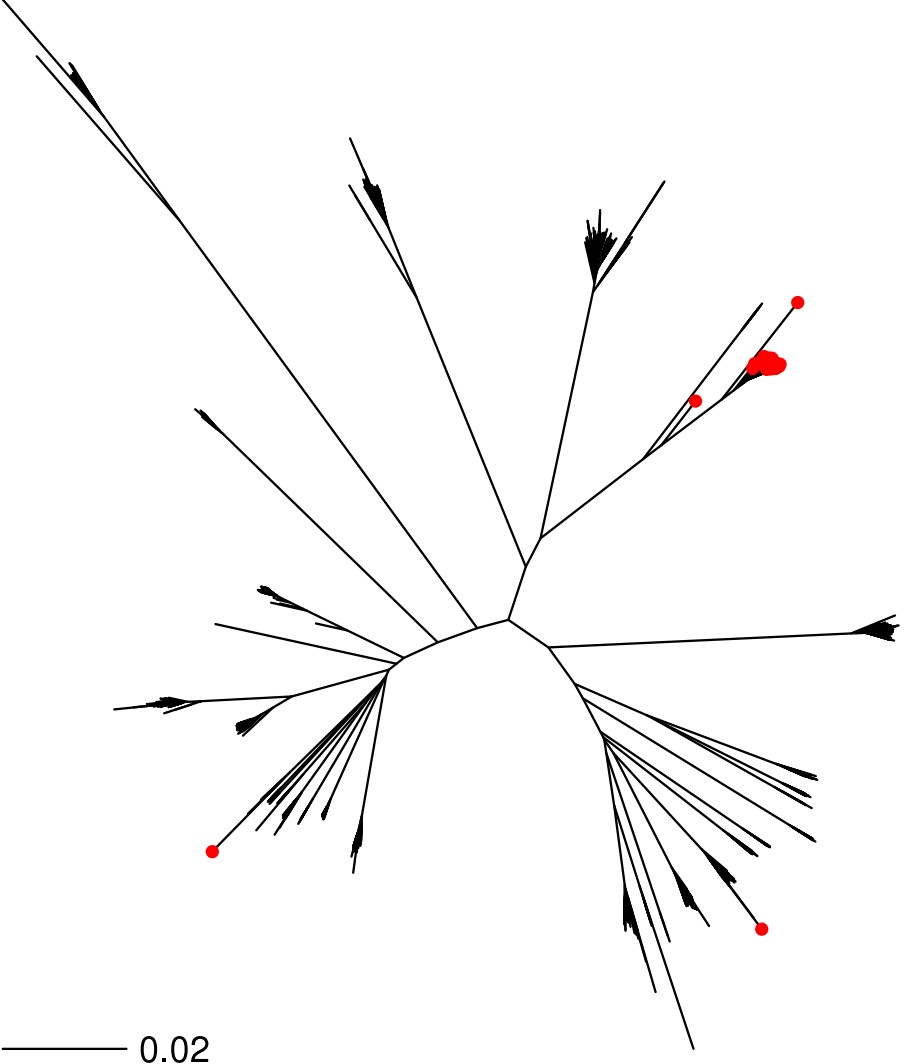

0.02

**Figure 6  Phylogenetic placement of SNP responsible for calling CC398 in HMP samples.** Strains from the whole genome phylogeny (Fig. 3) containing the synonymous SNP in the RNA are marked with dots. This figure shows that this SNP is found primarily in strains of the CC398 subtype, but there are two outlying strains that also carry the mutation.

having health insurance or not ($p$-value $= 0.0525$) and BMI ($p$-value $= 0.0276$) were predictors for the detection/presence of *S. aureus*, whereas for the multinomial logistic regression model with 4 outcomes, only main site ($p$-value $\leq 0.001$) and BMI ($p$-value $= 0.0251$) were predictors of the presence of *S. aureus* (Table 4). The estimated odds ratio for detecting the presence of any *S. aureus* subtypes in the airways compared to the oral cavity was 3.3 (95% CI [2.2–5.0]). This is consistent with our study and other previous studies showing that *S. aureus* is highly enriched in the anterior nares (nose) compared to any other body sites, and also body site could be a strong predictor for the presence of *S. aureus*. In the multinomial model where specific subtypes were examined, odds of detection of CC8, CC30 and other subtypes were all significantly elevated in the airways compared to the oral cavity (Table 4). Similarly, the odds of detecting any *S. aureus* subtypes in subjects

**Table 3  Within-body site Hamming score similarity compared scores generated from 10,000 random permutations of all body-sites.**

| Body site | Number of samples (minimum 4) | p-value |
|---|---|---|
| Anterior nares | 67 | 0.50 |
| Buccal mucosa | 54 | 0.0008 |
| Left retroauricular crease | 23 | 0.85 |
| Palatine tonsils | 6 | 0.62 |
| Posterior fornix | 9 | 0.007 |
| Right retroauricular crease | 28 | 0.99 |
| Stool | 7 | 0.035 |
| Supragingival plaque | 37 | 0.0008 |
| Tongue dorsum | 82 | 0.78 |

with higher BMI was 70% higher when compared to subjects with normal BMI. In the multinomial model, odds of detection of CC8, CC30 and other subtypes were all elevated for higher vs. normal BMI, but the higher odds of detection was more pronounced for the other subtype group (OR CC8 = 1.4, 95% CI [0.6–3.0]; OR CC30 = 1.1, 95% CI [0.5–2.2]; OR other subtype = 2.4, 95% CI [1.3–4.5]). Also the odds of detecting CC8 subtype tended to be higher in high BMI subjects while CC30 subtypes appeared to be associated with lower BMI. In the binary outcome model, subjects without health insurance had less detection of *S. aureus* compared to subjects with health insurance, with an estimated 50% lower odds of detection of any *S. aureus* subtypes among the uninsured. Even though race and ethnicity overall was not a statistically significant predictor ($p$-value = 0.28) for the detection of *S. aureus*, there was some indication that the odds of identifying any *S. aureus* subtype was higher among Hispanics compared to Non-Hispanic whites, with the odds ratio most elevated for detection of CC8 in the multinomial model (Table 4). However, we note that these odds ratios were based on only 45 samples from Hispanics included in our analysis (13 with detection of any *S. aureus*). Our analysis also confirmed previous results that gender was not a significant predictor for the detection of any *S. aureus* subtypes ($p$-value = 0.77). The odds of detecting *S. aureus* in females were 10% less than in males (Table 4). Age, breast fed or not, tobacco use (also previously identified in *Liu et al., 2015*) and history of surgery were not significant predictors of association of the presence of *S. aureus* subtypes in any human body sites ($p$-value > 0.1). Even though CC398 was enriched at the tongue dorsum we could not find any statistically significant association with the presence of CC398 in the tongue dorsum and eating a diet that contains meat.

A major limitation of our epidemiologic analysis was the imprecision in our estimated odds ratios for detection of *S. aureus* driven mainly by the small number of study subjects and to some extent by the homogeneity of the HMP population for factors such as age, race, tobacco use, health insurance, diet, and history of surgery (see Table S3). Despite this, we did observe evidence for more detection of *S. aureus* among subjects with higher BMI compared to those with normal BMI, and suggestion towards lower detection of *S. aureus* among subjects without health insurance.

**Table 4** Estimated odds ratios with 95% confidence interval for models with binary outcome as well as multinomial outcomes.

| Exposure variable | *Staphylococcus aureus* Present/not present OR (95% CI) | *p*-value | Presence of *Staphylococcus aureus* CC type | | | |
|---|---|---|---|---|---|---|
| | | | CC8 OR (95% CI) | CC30 OR (95% CI) | Other CC types OR (95% CI) | *p*-value |
| **Main body site** | | <0.0001 | | | | <0.0001 |
| Airways vs. Oral | 3.3 (2.2–5.0) | | 2.7 (1.3–5.4) | 2.5 (1.3–5.8) | 4.6 (2.7–7.7) | |
| Airways vs. Skin | 0.1 (0.0–0.3) | | 0.2 (0.0–0.8) | 0.1 (0.0–0.6) | 0.0 (0.0–0.2) | |
| Oral vs. Skin | 0.0 (0.0–0.0) | | 0.0 (0.0–0.3) | 0.0 (0.0–0.2) | 0.0 (0.0–0.1) | |
| **Diet** | | 0.4675 | | | | 0.3228 |
| Meat/fish/poultry at least 3 days per week | 1.4 (0.6–3.0) | | 0.4 (0.1–1.5) | 2.0 (0.4–10.4) | 4.0 (0.7–21.0) | |
| Meat/fish/poultry at least 1 day but not more than 2 days per week | 2.3 (0.6–8.9) | | 1.6 (0.1–15.9) | 3.1 (0.3–31.9) | 5.4 (0.5–62.5) | |
| Eggs/cheese/other dairy products, but no meat/fish/poultry (Reference) | 1 | | 1 | 1 | 1 | |
| **Gender** | | 0.6434 | | | | 0.776 |
| Male (Reference) | 1 | | 1 | 1 | 1 | |
| Female | 0.9 (0.6-1.3) | | 1.0 (0.5–2.1) | 0.7 (0.4–1.4) | 0.9 (0.5–1.6) | |
| **Age** | | 0.7197 | | | | 0.8055 |
| 3 years of age difference | 1.0 (0.9–1.1) | | 1.1 (0.9–1.3) | 1.1 (0.9–1.3) | 1.0 (0.8–1.1) | |
| **Breast fed or not** | | 0.6527 | | | | 0.8913 |
| Yes (reference) | 1 | | 1 | 1 | 1 | |
| No | 1.0 (0.7–1.6) | | 0.7 (0.3–1.8) | 1.0 (0.5–2.1) | 1.3 (0.7–2.5) | |
| Don't know/remember | 0.8 (0.5–1.3) | | 0.6 (0.2–1.6) | 0.9 (0.4–1.9) | 0.9 (0.4–1.9) | |
| **Tobacco use** | | 0.7522 | | | | 0.7173 |
| Yes | 0.9 (0.4–1.8) | | 1.2 (0.3–5.4) | 1.4 (0.5–3.8) | 0.6 (0.2–2.0) | |
| No (Reference) | 1 | | 1 | 1 | 1 | |
| **Have health insurance or not** | | 0.0525 | | | | 0.403 |
| Yes (Reference) | 1 | | 1 | 1 | 1 | |
| No | 0.5 (0.2–1.0) | | 0.0 (0.0–13.0) | 0.9 (0.3–2.6) | 0.5 (0.2–1.5) | |
| **BMI** | | 0.0276 | | | | 0.0251 |
| <22 | 1.1 (0.7–1.9) | | 0.4 (0.1–1.1) | 1.6 (0.6–3.8) | 1.7 (0.7–4.0) | |
| 22–25 | 1 | | 1 | 1 | 1 | |
| >25 | 1.7 (1.1–2.5) | | 1.4 (0.6–3.0) | 1.1 (0.5–2.2) | 2.4 (1.3–4.5) | |
| **RACE** | | 0.2815 | | | | 0.2791 |
| Hispanic | 2.0 (0.9–4.1) | | 4.7 (1.2–18.9) | 0.5 (0.1–2.8) | 2.3 (0.7–6.9) | |
| Asian | 1.3 (0.7–2.3) | | 2.6 (1.0–7.3) | 1.1 (0.4–3.2) | 1.1 (0.4–3.0) | |
| Black | 1.2 (0.5–2.5) | | 1.0 (0.2–5.6) | 1.9 (0.6–6.3) | 0.9 (0.2–3.1) | |
| White (Reference) | 1 | | 1 | 1 | 1 | |
| **Whether undergone any type of surgery** | | 0.9241 | | | | |
| Yes | 1.0 (0.4–2.5) | | 2.6 (0.4–18.7) | 0.0 (0.0–172.8) | 1.6 (0.4–6.5) | 0.5205 |
| No (Reference) | 1 | | | | | |
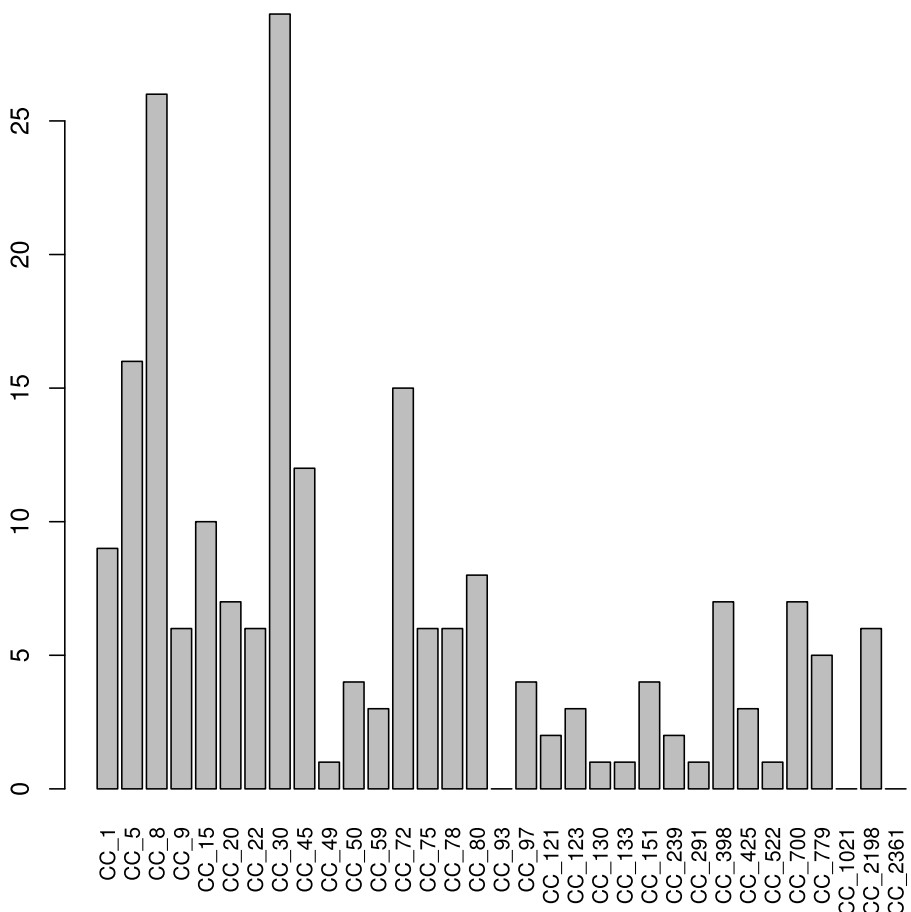

**Figure 7** *S. aureus* **subtypes in NYC subway samples.** 149 samples from the NYC subway with a *S. aureus* core coverage > 0.025X project were classified using binstrain with the v2 matrix. The figure shows counts of the number of samples with each subtype present with beta > 0.2.

## Geographical distribution of *S. aureus* subtypes found in the New York City subway setting

*Afshinnekoo et al. (2015)* sampled 1,457 surfaces at all 466 open subway stations in NYC between mid 2013 to February 2014 and shotgun sequenced them using Illumina technology, while at the same time collecting detailed metadata. We used our subtyping methodology to investigate *S. aureus* diversity within the study set, finding 149 of the metagenome samples to contain > 0.025X coverage of the *S. aureus* core genome. CC30 (29(20%)) and CC8 (26(19%)), respectively, were again the most common subtypes (Fig. 7). CC5 (17(11%)) and CC45 (12(8%)) were also common, as we found in the HMP data, but CC2198 ((4(10%)%)) and CC1 (9(6%)) were more prevalent than HMP.

The NYC metadata allowed us to ask whether there was any geographical structure to the *S. aureus* population that might reflect compartmentalisation of the city based on ethnicity or other reasons but we found little evidence of clustering. Sites containing *S. aureus* were distributed more or less randomly over the New York City map, as were individual subtypes (Fig. 8). Of the six most common subtypes, there was only one case (CC8) where the mean

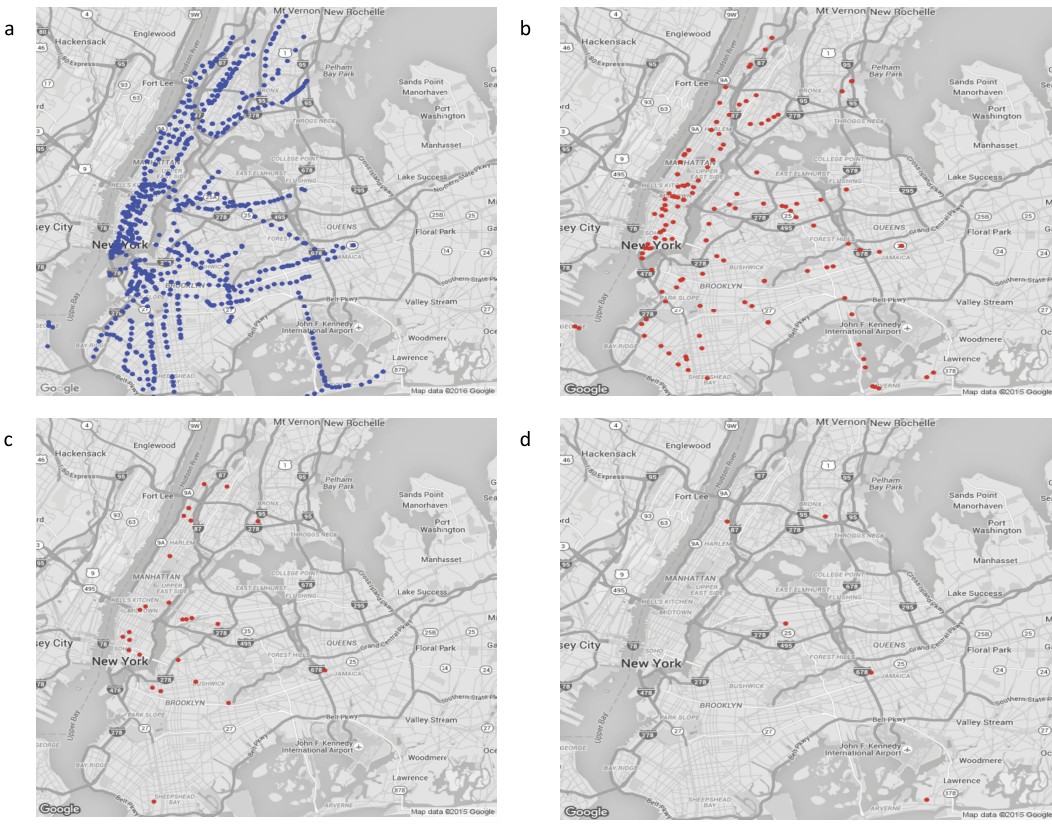

**Figure 8** **Distribution of *S. aureus* subtypes identified from the New York City subway metagenome data.** (A) A map of all sites sampled by *Afshinnekoo et al. (2015)* (31) in the central Manhattan area, (B) sites containing any *S. aureus* reads (C) sites containing CC30 (beta > 0.2; v2 classification), (D) sites containing CC22. Maps © Google.

geographical distance between sites was in the lower 5% of NYC sites sampled at random (10,000 permutations). There was a significant regression between Hamming distance between sites and the straight line geographic distance ($p < 0.001$) but the effect size was small ($R^2 < 1\%$). These result is in line with a recent study by *Uhlemann et al. (2014)*, who found there was no strong geographic signal in the phylogeny of 387 strains of USA300 (CC8-ST8) isolated in households in Manhattan. These results suggest movement of *S. aureus* subtypes around the city may occur over a shorter timescale than the movement of the human population.

## DISCUSSION AND CONCLUSIONS

In this work we developed a test to estimate the subtype profile of an important bacterial pathogen (*S. aureus*) from raw metagenomic data. We showed different mixtures of subtypes in different human subjects, and body sites and also gained an overview of the the biogeography of *S. aureus* subtypes from environmental metagenomic data. The *binstrain* method could be used in studies seeking to examine variation in other bacterial species.

*binstrain* explicitly models the presence of mixture of bacterial subtypes in the metagenome sample. The output is the assignment of metagenome data into bins that are

convenient for microbiologists to interpret. The sensitivity of *binstrain* assigning subtypes (in terms of accurate assignment at low genome coverage) is set by the amount of nucleotide variability in the species and the number of subtypes. It can be expected that in many cases, as we showed for the two datasets here, there will be less than $2\times$ genome coverage of the target species in the metagenome sample. The distinction between assignment to individual reference "strains" and subtypes, which represent populations of strains, is important. The set of reference strains of a bacterial species in public databases is usually biased towards a small number of genotypes (often high-consequence pathogens). This can lead to an uneven picture of the actual composition of the species when using reference-based assignment.

Here, we were able to construct population-based assignment of *S. aureus* by drawing on the unusually large number of diverse genomes sequences available in the public domain. Nevertheless, the v2 subtyping schema still leaves room for future improvement. The largest assignment problem was that only 97/157 (62%) ST5 strains were classified as within the the CC5 subtype (Table S2). This is being partly explained by variability within the ST5 strains themselves, presumably because of recombination: 36 ST5 strains were placed outside the main cluster of strains in the global *S. aureus* phylogeny. Future iterations of the SNP matrix could be improved by incorporating information for multiple reference strains of each genotype.

In this study we found evidence that the *S. aureus* subtype composition of different body sites in the same individual was more similar than would be expected from comparing random samples of the population. The result is in line with the hypothesis that *S. aureus* spreads between body sites of the same individual (*Kluytmans, Van Belkum & Verbrugh, 1997*).

The evidence for association of subtypes with different body sites was less conclusive. Despite the significant PERMANOVA *p*-value, the result was confounded by large differences in the variance of subtype composition between body sites. Permutation analysis (Table 3) showed that some body sites appeared to have more similar composition than expected by chance Body site specificity might arise from genetic adaptation to growth in particular human niches or from unequal distribution of subtypes in environments that serve as entry points to the human. For example, the striking finding that 90% of CC_398, a subtype commonly but not exclusively associated with livestock (*Lewis et al., 2008*; *Price et al., 2012*) was located in the tongue dorsum, may be because of ingestion of food.

Of the epidemiologic variables only body mass index (BMI) > 25 and possession of health insurance were associated with the presence of *S. aureus*. The subjects chosen for the HMP study were fairly homogenous in terms of age, ethnicity (*Ding & Schloss, 2014*), and absence of medical conditions, leaving little power to associate with conditions more prevalent in the general population. We did not adjust for multiple comparisons and it is always possible that observed associations are due to chance. However, we also note the limited power we had in this HMP population. The health insurance *p*-value (0.0525) corresponds to a 50% lower risk of carriage among the uninsured, which if true, would be a nontrivial effect size if high BMI and health insurance are in truth risks for *S. aureus* presence, these relationships may be connected to health factors outside those collected directly. For example, BMI may be associated with diet, exercise or other behaviors rather

than directly associated to *S. aureus* carriage; likewise the lower detection among uninsured people may reflect less contact with the medical system or socioeconomic factors. There were no strong links to the carriage of the two major subtypes, CC30 and CC8, and there was no geographical distinction between the two in New York City subway station samples. Understanding the reasons behind the distributions of *S. aureus* subtypes will take larger data sets.

This study illustrated how shotgun metagenomic datasets associated with metadata and made available in the public domain are a rich resource that can be mined repeatedly for results beyond the original purposes for which they were created. Given the current growth in metagenomic sequence production, these could soon number in the hundreds of thousands to millions of metagenome data sets. *S. aureus* subtyping can be performed on all of these data. Large-scale analysis subtype would give a much richer insight into *S. aureus* biogeography and colonization preferences than we have been able to achieve through conventional microbial sampling alone, a method which has been traditionally focused on one body site (anterior nares) and rarely takes into account the within sample genetic mixture of the species (*Albrecht et al., 2015*). The *binstrain* method, and/or alternative approaches described above, can be extended to species other than *S. aureus*, giving us more tools to explore the bacterial diversity of this planet.

## ACKNOWLEDGEMENTS

Part of this work was presented as a Master's thesis at Rollins School of Public Health, Emory University by SJJ towards his MPH degree in Applied Epidemiology. Access to de-identified data was approved under dbGAP agreement #18287-4. "Support for the development of NIH Human Microbiome Project - Core Microbiome Sampling Protocol A (HMP-A) was provided by the NIH Roadmap for Medical Research. Clinical data from this study were jointly produced by the Baylor College of Medicine and the Washington University School of Medicine. Sequencing data were produced by the Baylor College of Medicine Human Genome Sequencing Center, The Broad Institute, the Genome Center at Washington University, and the J. Craig Venter Institute. These data were submitted by the EMMES Corporation, which serves as the clinical data collection site for the HMP.

### Funding

This work was funded through development funds from Emory University School of Medicine. "Funding support for the development of NIH Human Microbiome Project—Core Microbiome Sampling Protocol A (HMP-A) was provided by the NIH Roadmap for Medical Research. The funders had no role in study design, data collection and analysis, decision to publish, or preparation of the manuscript.

### Grant Disclosures

The following grant information was disclosed by the authors:

Emory University School of Medicine.

NIH Roadmap for Medical Research.

## Competing Interests

Timothy D. Read is an uncompensated member of the Scientific Advisory Board of the metaSUB consortium and an Academic Editor for PeerJ.

## Author Contributions

- Sandeep J. Joseph and Timothy D. Read conceived and designed the experiments, performed the experiments, analyzed the data, wrote the paper, prepared figures and/or tables, reviewed drafts of the paper.
- Ben Li, Robert A. Petit III and Zhaohui S. Qin contributed reagents/materials/analysis tools, reviewed drafts of the paper.
- Lyndsey Darrow analyzed the data, reviewed drafts of the paper.

## Data Availability

https://github.com/Read-Lab-Confederation/staph_metagenome_subtypes.

Clinical data from this study were jointly produced by the Baylor College of Medicine and the Washington University School of Medicine. Sequencing data were produced by the Baylor College of Medicine Human Genome Sequencing Center, The Broad Institute, the Genome Center at Washington University, and the J. Craig Venter Institute. These data were submitted by the EMMES Corporation, which serves as the clinical data collection site for the HMP.

## Supplemental Information

Supplemental information for this article can be found online at http://dx.doi.org/10.7717/peerj.2571#supplemental-information.

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
