# Peer review of "The single-species metagenome: subtyping Staphylococcus aureus core genome sequences from shotgun metagenomic data"

_PeerJ, doi:10.7717/peerj.2571_

## Round 0.1 · original submission · Major Revisions

· Academic Editor

Major Revisions

I have now received two reviews of your paper and both reviewers found it very interesting and appropriate for the journal. However, both have found a number of issues, especially around justifications for approaches taken and further details of some of the methodology. They both make a number of additional suggestions. I find their comments informative and think adjustments to accommodate these concerns will help improve your paper. Therefore, I am recommending major revision.

·

Basic reporting

This manuscript describes the typing of S. aureus directly from public metagenomic datasets with binstrain. Perhaps the most impressive finding is the ability of the method to distinguish mixed genotypes at very low read depths, which is a known limitation.

I appreciate how the authors made files available on github for running their analyses. After scanning the documents, I think that more information on how to set up a binstrain run would be greatly appreciated. I couldn't find information on what the SNP matrix might look like, but is vital for binstrain to work correctly. A walkthrough of input types and analyses would help others set up their own analyses to test the binstrain method on other organisms.

Experimental design

I had difficultly following the validation experiments conducted at the beginning of the results. For example, MLST typing was performed on SRA genomes, but the resolution of subtypes was too low using matrix v1. How was that assessed? What forms a subtype and how does that relate to ST? Later on in the subway discussion, clonal complexes are referred to as subtypes.

The ability of binstrain to resolve mixtures at low coverages is remarkable given that many regions of the core genome are not informative for defining subtypes. Can you firmly define a lower bound on how much coverage is required for accurate typing? How many SNPs need to be identified for accurate subtyping?

Validity of the findings

Why did you not adjust for multiple comparisons in the associations? If you did, how would this affect the BMI and insurance associations? Would they no longer be significant?

The SNP matrix is important, but details on how the matrix was generated are missing. For example, how are duplicated regions handled? How about the effect of homoplasy in the matrix on accurate assignments?

Additional comments

Specific comments:

L70. Not sure what this means. Reference 2 uses a reference set of genomic information to classify unknowns. ConStrains uses a reference set of genomic data to perform classifications. This sentence should be re-considered, or reworded, in light of these results.
L76: The colon here appears to be incorrectly used and should probably be broken up into separate sentences
L94: "were" implies that these factors are no longer problematic, which I don't believe to be the case
L121: Change "short" to "sequence"
L123: Please clarify what you mean by "positive result"
L142: The Staph epi results are not in Figure 1, correct? This sentence makes it seem like Figure 1 will help confirm that binstrain was not affected by using an outgroup.
L145: Change metagenome to metagenomes
L185: You mention that >1 reads were mapped. Are any of your classifications actually based on only a single read mapping?
L360: Clarify the meaning of "are more directly to"
L385: I looked and couldn't identify the set of SNPs used for your classification matrix. This matrix should be included as supplementary data so others could reproduce your results
L432: How was the read downsampling performed?
L456: The methods to describe the conversion of reads to a multi-FASTA are too vague and should be expanded.

·

Basic reporting

This is a very interesting manuscript I enjoyed reviewing and worth publication at PeerJ with ability to draw a lot of readership value. However, I think the subtyping scheme has opportunity to further improve. The scheme also needs to demonstrate its stability when diverse S. aurues will be include. Recommendations were made at general comment sections

Experimental design

The strain sets selected for those limited 40 subtypes where based on limited number of reference genomes and the choice of the genomes were driven by traditional Clonal complex's. When NCBI has close to 7000 genomes available, robustness of such subtyping is not beyond question? As authors are using >100,000 known S. aureus SNP, why not start subtyping the isolates based on SNP first. I’ll strongly suggest considering such approach to develop a robust population structure.

Validity of the findings

No comments

Additional comments

This is a very interesting paper, with promise of making significant advancement in rapid strain sub-typing capability of S. aureus. Whereas author put forward tremendous effort in developing this subtyping scheme, I have some major concern and recommendations for consideration:

(1) I am particularly not clear about the rational behind using Mauve and fineSTRUCTURE to identify discrete population structure of S. aurues, based on the core portion of the chromosome, particularly knowing the fact that S. aureus have an evolving pan-genome, and previous studies (Hall et a., 2010, Jamrozy et al., 2016) have already documented both core genome heterogeneity, which often is much less than in the smaller fraction of the accessory genomes, which frequently demonstrates high degree of temporal variation. IMHO, the study should start with high-resolution SNP phylogeny and discrete population structure should be identified based of the SNP phylogeny. As authors are using >100,000 known S. aureus SNP, why not start subtyping the isolates based on SNP first. I’ll strongly suggest considering such approach.
(2) Another key question was the detection of only 40 subtypes, which is even smaller than MLST types available today which lacks sufficient resolution to distinguish among many S. aureus strains. Do authors think the gamma –diversity fo S. aureus could be explained by 40 subtypes? Even based on Fig.S2 you documented about 56 MLST types? What’s the explanation behind even limited number of subtypes using much higher resolution typing?
(3) Furthermore, what is the stability of these 40 subtypes, authors has not shown any data demonstrating these population structures are pretty robust and stable. My concern is, when you started with 43 completed genomes 19 subtypes were detected, than when additional 21 unfinished genome projects from diverse CCs were added the subtype reached to 40, IMHO, this is a poor way to establish the fundamental part of the subtyping scheme. What will happen when a bunch of genomes from NCBI (which has about ~7000 genomes now) added to this? Particularly, I request to include 24 distinct genome groups (http://www.ncbi.nlm.nih.gov/genome/?term=Staphylococcus+aureus ) to demonstrate the stability of the these subtyping scheme.
(4) What was the rational in choosing 0.025X coverage when false negative rate was highest at low coverage and 91% assignment was achieved at 0.5X coverage, why not start with 0.5X .
(5) Higher misclassification observed for CC5 might be due to fact that initial subtyping was heavily influenced by the choice of additional genomes belonging to different CC’s, and the known fact that ST5 complex has at least a dozen of ST’s like ST_5, ST_85, ST_105, ST_125, ST_225, ST_634, ST_764, ST_1178, ST_1447, ST_371, ST_228, ST_231. Do you r scheme has distinct subtype for each of these 12 ST’s?

Finally, this is truly a beautiful piece of work and much needed one for the field. Implementation of these subtyping scheme on HMP and NY Subway datasets have already shown some very interesting results. Subsequent improvement of such scheme may go beyond to this answering some additional question in the population biology of S. aureus.

---

## Round 0.2 · Minor Revisions

· Academic Editor

Minor Revisions

Just one more issue to deal with. Then give the whole thing a good once over for final submission. I will not send it out for review again. Thanks for accommodating the reviewers' critiques so well.

·

Basic reporting

The authors have done a good job at addressing my previous concerns

Experimental design

No Comments

Validity of the findings

No Comments

Additional comments

L77: Not sure what is meant by "with the metagenome same", please edit as necessary

---

## Round 0.3 · accepted · Accept

· Academic Editor

Accept

Thanks for that last revision!